# Cockle Shell-Derived Calcium Carbonate (Aragonite) Nanoparticles: A Dynamite to Nanomedicine

**Maryam Muhammad Mailafiya** [1,2] iD**, Kabeer Abubakar** [1,2] iD**, Abubakar Danmaigoro** [3] iD**, Samaila Musa Chiroma** [1,4] iD**, Ezamin Bin Abdul Rahim** [5]**, Mohamad Aris Mohd Moklas** [1,*] iD **and Zuki Abu Bakar Zakaria** [6]

1    Department of Human Anatomy, Faculty of Medicine and Health Sciences, University Putra Malaysia, Serdang 43400, Selangor Darul Ehsan, Malaysia
2    Department of Human Anatomy, College of Medical Sciences, Federal University Lafia, Lafia, Akunza 950101, Nasarawa State, Nigeria
3    Department of Veterinary Anatomy, Faculty of Veterinary Medicine, Usman Danfodiyo University, Sultan Abubakar 840213, Sokoto State, Nigeria
4    Department of Human Anatomy, Faculty of Basic Medical Sciences, University of Maiduguri, Maiduguri 600230, Borno State, Nigeria
5    Department of Radiology, Faculty of Medicine and Health Sciences, University Putra Malaysia, Serdang 43400, Selangor Darul Ehsan, Malaysia
6    Department of Preclinical Sciences Faculty of Veterinary Medicine, University Putra Malaysia, Serdang 43400, Selangor Darul Ehsan, Malaysia
*    Correspondence: aris@upm.edu.my; Tel.: +60-193387042

**Abstract:** Cockle shell is an external covering of small, salt water edible clams (*Anadara granosa*) that dwells in coastal area. This abundant biomaterial is hard, cheap and readily available with high content of calcium carbonate in aragonite polymorphic form. At present, cockle shell-derived calcium carbonate nanoparticles ($CSCaCO_3NPs$) with dual applications has remarkably drawn significant attention of researchers in nanotechnology as a nanocarrier for delivery of different categories of drugs and as bone scaffold due to its beneficial potentials such as biocompatibility, osteoconductivity, pH sensitivity, slow biodegradation, hydrophilic nature and a wide safety margin. In addition, $CSCaCO_3NP$ possesses structural porosity, a large surface area and functional group endings for electrostatic ion bonds with high loading capacity. Thus, it maintains great potential in the drug delivery system and a large number of biomedical utilisations. The pioneering researchers adopted a non-hazardous top-down method for the synthesis of $CSCaCO_3NP$ with subsequent improvements that led to the better spherical diameter size obtained recently which is suitable for drug delivery. The method is therefore a simple, low cost and environmentally friendly, which involves little procedural steps without stringent temperature management and expensive hazardous chemicals or any carbonation methods. This paper presents a review on a few different types of nanoparticles with emphasis on the versatile most recent advancements and achievements on the synthesis and developments of $CSCaCO_3NP$ aragonite with its applications as a nanocarrier for drug delivery in nanomedicine.

**Keywords:** cockle shell; calcium carbonate; nanoparticles; nanomedicine; aragonite; biocompatibility; drug delivery

## 1. Introduction

Recent developments in advanced active delivery systems for drugs, elicited by nanotechnology had led to their application in modern science [1–3]. Conventional free-drug delivery through

subcutaneous, oral, intramuscular or intravenous routes, has led to drug biodistribution in the body via blood capillaries and accumulation to a certain concentration at the target site to exert therapeutic effects [2]. However, there is a paucity of the desirable effects of these free drugs due to poor bioavailability as a result of insolubility, hydrophobic nature, poor absorption and rapid metabolism [3]. In the quest to overcome some limitations associated with these therapeutic agents, the fast-growing field of nanotechnology focuses on diverse exploratory ways for researchers in the field of biomedical and pharmaceutical sciences. Advanced nanotechnology has paved ways for the use of convenient, affordable and noncomplex methods for synthesizing different nanoparticles from a range of abundant natural biomaterials and complex organic materials for industrial and medicinal purposes [4].

Nanoparticles are minute ultrafine particles with dimensions ranging between 1–1000 nm (usually 5–350 nm in diameter). They can be produced using different types of biocompatible materials [5,6]. The use of nanoparticle-based drug carrier systems in the field of nanotechnology has become an area with a novel attention in the past few decades due to their unique and superior properties that enable functionalisation at both molecular and cellular levels [7–14]. Advanced novel drug delivery systems help to boost the efficacy of therapeutic drugs as well as minimising the rate of a drug's off targeted effects thereby preventing cytotoxicity to normal cells [8]. This brought forth a useful enhanced therapeutic efficacy through suitable modifications of a drug's bioavailability, pharmacokinetics and serum stability [9] as well as specificity in drug release which is elicited by response to a particular stimulus such as ultrasound intensity, pH, magnetism and temperature [10].

Cockle (*Anadara granosa*) with a Malaysian native name "kerang" belongs to the family of *Cardiidae*, which is a small, salt water edible clam that is popularly referred to as marine bivalve mollusc [11]. It is native to coastal regions of South East Asia (Malaysia, Thailand and Indonesia). These important sea species mostly dwells in coastal area and it is a common important source of calcium carbonate ($CaCO_3$) with abundant biomaterials for biomedical purposes [12]. It has tremendous striking properties, also it is cheap and readily available with abundant high quality and pure $CaCO_3$ in aragonite polymorphic form, which is used in drug delivery [4].

$CaCO_3$ is an inorganic calcium salt originated from varieties of shelled molluscs, limestone, coccolithophores, plant ashes, chalk and marble which is recently being studied in the field of nanotechnology as one of the potential porous biocompatible and pH sensitive material [6,11]. Further, its solubility has been stated to be exponentially and inversely proportional to its pH [10,13]. Its physicochemical properties are easily regulated as well as surface morphological chemistry and method of production [14,15]. $CaCO_3$ is one of the emerging inorganic nanoparticles which exists in three different polymorphs such as vaterite, aragonite, and calcites [15]. It also possess a peculiar property of low thermodynamic stability thus, its size and shape are in concordance with the method and conditions of laboratory preparations [11]. In addition, numerous studies in recent years, were carried out on its toxicity to prove its wide safety margin both *in vivo* and *in vitro* [10,16–20]

A significant number of publications have appeared in recent years documenting the application of CSCaCO₃NP of aragonite polymorph nanoparticles on different models of animals and cell lines for various therapeutic purposes with regard to their synthesis, functionalization, and both *in vitro* and *in vivo* toxicity studies [4,15,16,19,21,22]. Among all available documented literature on CSCaCO₃NP with unique features so far, no review has been documented on the overview of this very nanoparticle of cockle shell biogenic origin. Hence, this current review highlights a few of the varieties of nanoparticle with some versatile advances on the synthesis and developments of CSCaCO₃NP as a nanocarrier in drug delivery systems.

## 2. Nanotechnology and Nanomedicine

Over the past few decades, nanotechnology has gained tremendous attention with good future prospects focusing more on nanoparticles, which is basically the bedrock of nanotechnology [1,13,17, 23–26]. Nanotechnology is an important aspect of science that focuses on the continuous designing and manipulation of materials (atoms and molecules) to produce structures at the nanometre scale size

ranging from smaller nanometre to 100 nm with unaltered initial unique features of the material used, with broad nanoscale schemes in clinical applications for therapeutic, protective as well as diagnostic purposes [8,27]. The entire application of nanotechnology from the synthesis processes, control release profiling, monitoring of biological processes and diagnosis is referred to as "Nano-medicine" [8]. This comprehensively means a process of transformation and encapsulation of a drug's molecules using nanostructures with or without the help of a carrier materials, masking some inherent drawbacks and limitations of free drugs such as poor bioavailability, hydrophobicity, high dosages, rapid assimilation, short half-life of photo degradation, poor selectivity as well as off targeted effects [28]. Conventional utilization of free drugs portrays poor bioavailability, low efficacy, non-selectivity and undesirable side effects [8,29]. For the past few years, a considerably large number of poorly soluble drug candidates has gradually increased as a result of the use of high-throughput screening and combinatory chemistry in drug discovery [30–32]. Approximately 70% of marketed drugs and many new chemical drug candidates, medicinal herbs as well as food supplements sometimes fails to be absorbed in the gastrointestinal tract (GIT) and often possess poor intravenous circulation and muscular tissue absorption after administration [33]. The low solubility nature of drugs limits their dissolution rate leading to a variety of issues which consequently results in low bioavailability as well as an erratic absorption pattern of drugs in biological systems [34]. However, an alternative way of overcoming such problems can be achieved by dose escalation although, this could result in undesirable effects associated with increased toxicity leading to patient's non-compliance [32]. Delivering therapeutic compound to the desirable localised site is quite challenging for the treatment of many ailments [29]. However, the application of nanomedicine have currently helped to overcome some of the problems of free dugs for therapeutic purpose [35]. This could be due to the uniqueness of the physicochemical properties presented by nanoparticles such as; (a) increased solubility; (b) increased drug pharmacokinetics; (c) co-delivery of multiple drugs to the same specific location at the same time (synergistic treatment); (d) enhanced bio distribution and bioavailability of drug to the targeted area; (e) improved drug permeability and retention effects; (f) increased specificity of drugs to targeted site of interest; (g) increased surface area to volume ratio and (h) decreased patient-to-patient variability [2,36].

## 2.1. Nanoparticles

Nanoparticles have unique physicochemical characteristics at the nanoscale compared to their former properties at micron or higher scales due to certain modified phenomenon like increased reactivity or stability in a chemical process, enhanced mechanical strength, enhanced solubility and relatively large surface area to volume ratio, which make them applicable in various fields of research studies [1]. Nanoparticles differ in physicochemical properties such as their surface morphology (uniform or irregular surface variations), some are amorphous or crystalline with loose or agglomerated features which are either single or multi crystal solids [37]. Nanoparticles have the ability to cross the brain barriers especially the blood brain barrier thereby shielding the encapsulated therapeutic agent and efficiently delivering them to the targeted areas, thus, it is considered suitable for drug delivery systems for the treatment of neurodegenerative diseases [38]. Nanomaterials are indeed very effective and have potential abilities to alter and modify a drug's mechanisms of action, bone reinforcement, DNA structure probes, tissue engineering, drug and gene delivery, protein detection and tumor destruction, molecular imaging, diagnostics, regenerative medicine, pharmaceutical sciences, engineering and industrial chemistry [39].

Types of Nanoparticles

Abundant biocompatible nanoparticles differ in chemical constituents, nature and physical properties. These nanoparticles are efficient enough to maintain a slow sustained release of loaded drugs at the targeted area which is vital for therapeutic purposes [9]. They are generally applicable in the biomedical field for drug delivery systems, thus, their drug carrying capacity, stability and

delivery, encapsulation or adsorption determines their level of effectiveness [40]. Thus, different types of nanoparticles are highlighted below:

(a)　Dendrimers

Dendrimers are branched nanosized polymers that have simple modifiable surfaces with numerous chain ends which can be adjusted to perform special chemical functions [41]. These macromolecules have monodispersed homogenous shapes, sizes and surfaces with three basic components: repetitive branching units, a terminal group and a central core promoting modified surface functionalities [42]. They are hydrophilic, non-immunogenic and biocompatible in nature which are used for drug and gene delivery [41]. Convergent growth and divergent methods are the well explored methods for synthesizing dendrimers [41,42]. The three major types of dendrimers are: poly-L-lysine (PLL), poly(propylene imine) (PPI) and polyamidoamine (PAMAM) [42]. The presence of functional groups at the extremist surface of dendrimers, promotes its conjugation with other moieties to aid delivery and detection of other diseases thus, drugs can bind to these functional groups or get completely entrapped at the intramolecular cavity of the dendrimers [41,43]. The biomolecule-like physicochemical properties, alongside their nanosize, globular shape, multivalence as well as their tubular inner cores, advances their importance thereby attracting interest of scientific researchers [44]. Toxicity and hemolytic properties of dendrimers set their major drawbacks leading to a concern regarding their safety [44]. The toxicity can be due to the characteristic nature of the dendrimers which is related to its core chemistry [41,44].

(b)　Micelles

Micelles are self-assemblies of co-polymers which are supramolecular aggregates of surfactant molecules dispersed in a solution colloid with a diameter ranging between 10–100 nm [45,46]. The small molecule that forms conventional micelles consists of a hydrophilic (polar or charge) "head" and hydrophobic "tail", which is composed of hydrocarbons of long fatty acid [47]. It comprises of a "core" and "shell" which forms the inner and outer domains respectively. Micelles offer an outstanding advantage in drug delivery especially via the oral route thus, protects oral administered insoluble drugs by delivering them to the desired site at the concentration exceeding its intrinsic water solubility [48–50]. The outer shell, also known as the corona, is the hydrophilic part of the polymer that protects drugs from inactivating in the biological environment while the hydrophobic core sequesters hydrophobic drugs [48,51]. Encapsulation of drugs within the core are largely dependent on the compatibility between drug molecules and the hydrophobic core which can be estimated by comparing the properties of the hydrophobic segments and that of the polarity of the poorly water soluble drugs [32]. The outer shell serves as a physical shield that stretches its projections outward to prevent micelle–protein or micelle–micelle interactions [46]. Although micelles are considered as delivery vehicles for numerous insoluble drugs [49], they suffer some significant setbacks, thus, limiting their usage as nanocarriers, especially on gene delivery, since they have no convenient method of sequestering and delivering RNA and DNA, and as a result of that, their potential use as a delivery system is hindered [47].

(c)　Liposomal Nanoparticles

Liposomes are spherical materials representing a perfect model of bio-membranes and cells [52]. They are biocompatible and biodegradable carrier systems. Their structural architecture consists of vesicles which are made up of one or more lipid bilayer with a diameter ranging from 0.02 to 10 μm or several micrometres [29,53,54]. They are the most old prominent and the most widely used nanocarriers that were discovered in the mid-1960s and presently attained their position in modern drug delivery, food industry and gene therapy [55]. The lipid bilayer consists of an aqueous core wrapped by synthetic or natural phospholipids especially phosphatidylcholine but may also include other lipids [50]. Based on the size, number and structure of the lipid bilayer, the liposome is classified into three classes viz; large unilamellar vesicles (LUV), small unilamellar vesicles (SUV) and multilamellar vesicles (MLVs). Encapsulation of hydrophilic, amphiphilic and hydrophobic drugs is done at the aqueous

core of the liposomes due to their flexibility hence, solubilisation of insoluble drugs occurs within the phospholipid bilayers [29,55]. They are abundant with enormous diverse properties such as distinct structures, size, flexibility, physicochemical compositions and variety of surface modifications which promotes them to be efficient carrier systems for both passive and active delivery [29]. They are widely used in food, cosmetics, agriculture and pharmaceuticals industries as a carrier system for safe delivery of materials like genes, drugs and nutraceuticals [52]. They are naturally occurring non-toxic biomaterials that are extremely versatile. Hence, their lipid composition can be manipulated thus, particle size alters alongside surface charge [54]. Some disadvantages of liposome carrier systems are associated with a low transportation rate in vivo as they are rapidly cleared from blood due to the phagocytic activities of the body cells [55]. They are sensitive to temperature, bilayer rigidity and pH. The process of encapsulation has a significant effect on the encapsulation efficiency of liposomes thus; liposomal-drug synthesis holds advantages when there are adequate amounts of drugs and lipids in proportion. However, excessive doses of lipids can interfere with expected pharmacokinetics of the liposomal drugs and sometimes can be toxic [56].

(d)    Chitosan Nanoparticles

Chitosan is a biodegradable, biocompatible and non-toxic polymeric material used in the synthesis of nanoparticles for drug delivery systems [57]. It is a linear polysaccharide mostly obtained from crustacean shells and fungi cells [58]. Chitosan is composed of N-acetylglucosamine and glucosamine derived from N-deacetylation of chitin which is the second most abundant natural bio-polymer after cellulose [59,60]. The excessive density of amine groups present at the back-bone of deacetylated chitosan promotes strong electrostatic interactions with genes and proteins [61]. The cationic natural polymer for the past few years have being broadly investigated particularly as the main material in synthesizing carriers for therapeutic proteins and as non-viral gene carrying vectors [61–63]. Chitosan nanoparticles have an advantage of controlling the release of a drug thereby increasing drug stability and decreasing toxicity [58]. They are capable of passing via the biological barriers *in vivo* to deliver drugs to the lesion site for enhanced therapeutic efficacy [64]. It has a wide range of biological importance such as an antimicrobial property, thus, has the ability to prevent the spread of infections [57], as a wound healing-accelerator which promotes coagulation by amplifying the functions of inflammatory cells [65]. In addition, chitosan can be used as bandages to prevent excessive bleeding during injury [65], as well as an antibacterial agent, that helps to aid the delivery of dermal drugs [64]. Unfortunately, it has poor solubility in water and organic solvents which make it difficult to be modified chemically [66,67].

(e)    Carbon Nano Tubes

These nanosystems consist of graphene nanofoil and a honeycomb lattice of carbon with a diameter range between 0.5–3 nm and a length between 20–1000 nm. They are crystalline allotropes of carbon with hollow cylindrical nanostructures that have a remarkable strength, stiffness, excellent electrical properties (sound thermal conductivity, semi conductivity or insulation) and mechanical properties [8,29]. These hollow core nanoparticles are generally nontoxic to humans and can sometimes be referred to as nano-capsules that are highly sensitive to electromagnetic and thermal radiation (heat and light) [43]. Their functionalization promotes easy penetration of the cell cytoplasm and nucleus, increased solubility as well as a good nanocarrier for peptide and gene delivery [29,55]. They are characterized into single walled nanotube (single layer) or multi-walled nanotube (multiple layer). The structures consist of hexagonal networks of carbon atoms arranged in a fashion like a graphite sheet rolled at specific and discrete angles forming a cylindrical shape [40]. Carbon nanotubes are applicable in gene silencing, vaccine delivery, gene delivery, cell internalization, cell specificity, diagnostics, drug delivery, peptides and nucleic acids transportation [55,68]. Unfortunately, the production of nanotubes is relatively expensive and so would be difficult to be implemented in modern day nanotechnology [69]. Further, they are extremely small which usually limits their use, in fact the mechanisms of how they work is yet to be understood by scientists [68,70]. In addition,

carbon nanotubes are documented to have associated toxic adverse effect in vitro to different types of cells such as; human embryonic kidney, rat brain neuronal cells and human keratinocytes cells [70]. Unmodified carbon nanotubes were documented to initiate the formation of lung granulomas in mice when administered intra-tracheally [71].

(f)　Metal-Organic Frameworks (MOFs)

MOFs are organic–inorganic hybrid materials with three different dimensional structural topologies (1D, 2D and 3D) that consist of organic ligands and inorganic metal ions [72]. Their distinct features such as micro/mesoporosity with ultrahigh surface area and potential chemical functionalisation promoted their wide applications in the biomedical field [62–64]. MOF nanoparticles generally have high loading capacity and controlled release properties which makes them excellent candidates for drug delivery [73]. The adjustment of the pore size and structural functional groups makes it advantageous over rigid nanocarriers [74]. MOFs are highly biocompatible with less or no toxic effect in vivo [75]. They are synthesized with crystal sizes ranging from a few nanometres to micrometres. The micrometre dimensional MOFs are widely used for gas purification, catalysis and storage [73]. However, the application of MOFs for drug delivery and sensing requires miniaturisation at a nanoscale range between 1–500 nm [76].

(g)　Gold Nanoparticles

Gold nanoparticles are biocompatible materials that consist of several tuneable monolayers, which provide easy complete control of surface properties for targeted delivery, stability and controlled release [77]. They are nanoparticles within the range of 1–8 µm which exhibit different shapes such as sub-octahedral, octahedral, multiple twined, decahedral, spherical, nanotriangles, nanorods, nanoprisms, irregular shape etc. [78]. They obtained a significant role in the field of clinical diagnosis, pharmaceuticals and biomedicine due to their unique physicochemical properties [79]. The properties of gold nanoparticles differed from its bulk form because unlike the bulk form the nano sized are wine red in solution which is antioxidant in nature [78]. Gold nanoparticles are less hazardous due to their comparative chemical stability [22]. In addition, they are biocompatible and they hardly interfere with other labelled biomarkers in the body [79]. They are used for the delivery of anticancer agents and other therapeutic agents for the treatment of measles, chicken pox etc. [78].

(h)　Metal Phenolic Nanoparticles (MPNs)

MPNs have a supramolecular coordination structure that consists of polyphenols and metal ions [80]. These materials can be coated on various substrates to form films of nanostructure and due to their excellent multifunctional properties, their applications have been extended to a wide spectrum of fields, such as drug delivery, engineering, biotechnology, food technology and bioimaging [81]. They are highly biocompatible with versatile functionalisation, pH-responsive disassembly and low toxic effect which allow enhanced intracellular drug release [82]. MPNs possess negatively charged surfaces with flexible films which make them take different shapes suitable for drug delivery [83].

(i)　Solid Lipid Nanoparticles

Solid lipid nanoparticles are a spherical nanomaterial with an average diameter ranging between 10–1000 nm. They consist of a surfactant stabilized solid lipid core matrix that aids solubilisation of lipophilic molecules [84]. The surfactant is used as an emulsifier at a concentration of 0.5% to 5% to promote stability thus, the physiologically tolerated lipid component remains solid at both ambient temperature and body temperature [85]. The quality of surfactants and lipids greatly affects the physicochemical properties of the nanoparticles [86]. They are the most recent and effective lipid base nanoparticles that hold good future prospects for oral delivery of poorly water soluble drugs [87]. Solid lipid nanoparticles are highly stable and demonstrate prolonged drug release with avoidance of organic solvent during the synthesis process which makes it safer than polymeric carriers [85]. It has excellent biocompatibility properties with application versatility, although they have a constrained

advantage in drug delivery due to their unpredictable gelation tendency and poor encapsulation efficiency as a result of the crystalline nature of solid lipid [88].

(j)    Inorganic Calcium Carbonate

Calcium carbonate ($CaCO_3$) is a naturally occurring inorganic biomaterial which is used in the production of nanoparticles for different therapeutic purposes [15]. It is a slow biodegradable solid white substance occurring as limestone, marble and the main component of snails, pearls, eggshells and marine corals, and is hygroscopic in nature [89,90]. It exists in three polymorphs: aragonite, vaterite and calcite among which calcite is the most thermodynamically stable form [11]. In addition, inorganic $CaCO_3$ possesses the desirable large specific surface area, which provides it with higher loading capacity as well as stimuli-responsive drug controlled release properties when compared to the organic nanocarriers such as; dendrimers, micelles and liposomes [72]. The aragonite polymorph has drawn tremendous research attention mainly due to its strong biocompatible nature and large surface area [91]. Synthesis variables such as pH of the solution, reaction time, temperature, ion concentration and ratio, concentration of additive and stirring time greatly affects the physicochemical properties and the nature of the polymorph [90]. Inorganic calcium carbonate nanoparticles for biomedical applications have undergone broad investigations which shows unique characteristics due to their dual applications as a nanocarrier for loading different categories of drugs and as bone substitution [6]. It is an essential material used extensively in various therapeutic applications such as a nanocarrier for gene delivery, antimicrobial agent, anticancer agent and antioxidant agent which draws a remarkable higher advantage over other nanocarriers especially due to its less toxic nature and cost effectiveness [92].

*2.2. Polymorphism of CaCO$_3$NP*

$CaCO_3$ exists in three different kinds of crystalline polymorphs which are anhydrous in nature, they are; aragonite (needle-like), calcite (rhomboidal) and vaterite (spherical) as shown in Figure 1 [4,93]. The three different polymorphs vary in stability ranging from the most stable form which is calcite to the metastable forms which are vaterite and aragonite, these two readily metamorphosise into the stable polymorph [9]. Although, vaterite readily absorbs moisture and it is highly unstable especially when it comes in contact with water, it dissolves slowly and changes to calcite form [93]. Nevertheless, the synthesis condition is directly proportional to the different morphological forms of $CaCO_3$, for examples, temperature, reactant concentration and nature of additives all affect its stability [94]. Vaterite crystals exist as hexagonal form and it naturally occurs in organic tissues, as urinary calculi and gallstones, mineral springs etc. [6,9]. The trigonal–rhombohedral form is the most common naturally occurring calcite crystal and the absolute rhombohedral calcite is uncommon as natural crystals [95]. However, due to its thermodynamic stability under ambient conditions, calcite polymorphs has been used extensively in the industry [96]. $CaCO_3$ in aragonite crystals form differs from calcite by its nature of occurrence, the aragonite exists in an orthorhombic system with crystals which results in the formation of pseudo-hexagonal crystals due to continuous twinning [9]. In addition, the aragonite polymorph is more dense than calcite and so can be unravelled, synthesized and replaced by bones [15] however, aragonite has been applicable in tissue engineering, bone repairs and anticancer drug carrier [39].

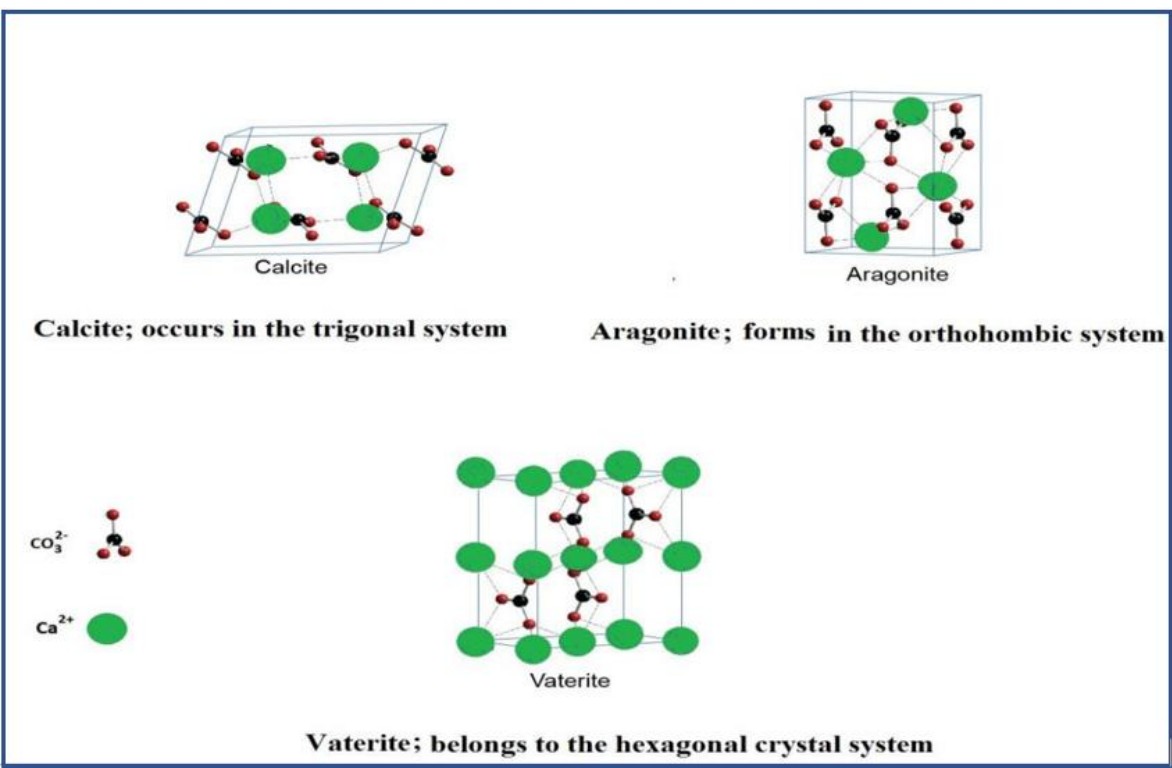

**Figure 1.** The three polymorphic forms of calcium carbonate extracted and modified from Dizaj et al. [6].

### 2.2.1. Aragonite

Aragonite is found naturally in almost all mollusc shells as well as the endoskeleton of cold and warm water corals [6]. The porous nature of the cockle shell polymorphic $CaCO_3$ nanocarrier derived from aragonite, enables it to incorporate large amounts of drugs thereby yielding a good encapsulation efficiency to boost the drug's bioavailability [17]. Aragonite is less stable than calcite and it is a high-pressure polymorph [91]. It is a virtually less thermodynamically stable, less available form of crystalline calcium carbonate developed and synthesized in the laboratories when compared to vaterites polymorph [16]. The surface chemistry (size and shape) of the aragonite nanoparticle produced rely solely on the conditions and method of preparations [13,16].

### 2.2.2. Advantages of Aragonite over Other Polymorphs

Among the three polymorphs, aragonite polymorph is the newest synthesized nanoparticles that emerged as one of the most popular targets by scientists for profound exploration in many areas, especially in bone tissue engineering, pharmaceutical and biomedical sciences [10,12,18,97,98]. Each of the different polymorphisms possesses different properties, which determine their special characteristics [99]. Aragonite among other polymorphs exhibits diverse medical applications due to its dual functions (as targeted drug delivery and as bone scaffold) [12,100]. Although, it is thermodynamically less stable than other polymorphs, it possesses superior density than calcite [12,98, 101]. It is more sensitive to temperature and pH changes [10,82,83]. Further, it is more advanced in surface functionalisation and possesses some morphological variations depending on the type of the method of synthesis [100–103]. Aragonite among others maintain high mechanical strength [98,101] and slow biodegradation promoting its sustained release performance [15,91,104].

### 2.3. Cockle Shell (Anadara Granosa)

Cockle shells are found in coastal area and are external coverings to the based bivalve shell mollusc [11]. They are found in the shallow tidal and low tidal areas of many enclosed bays, estuaries

and harbours [11,105]. Sea cockles are in abundance in the coastal area of Malaysia and its massive production increases yearly as it is mostly dumped in refuse in an open land or water bodies as untreated waste products with an unpleasant smell. Cockles are believed to be the most abundant sea species cultured in Malaysia [10]. The cheap sea shelled mollusc is common, readily available and often prepared and served as a local dish as a good source of proteins. In 2001, the aquaculture of cockle shells dominated about 4272.01 hectares of land with the output of 80,000 metric tons annually [12], however, as of 2007, the Malaysians were having 1055 farmers on 6000 hectares of land working on cockle agricultural cultivation thus, signifying how vast and high level of the waste cockle shells produced [89].

The biomass sea cockle shells (*Anadara granosa*), consist of about 98% calcium carbon elements which is useful for bone reformation and hence, it is considered as a potential pioneering material for the synthesis of biomaterials in orthopaedic medicine [12]. The remaining 1.3% of the total mineral composition consists of other useful elements such as; oxygen (O), sodium (Na), phosphorus (P), potassium (K), zinc (Zn) etc. [4]. This clearly explains why water corals and sea shells are liable and applicable in nanotechnology. The increase in the use of nanoparticles for biomedical applications encourages the continuous synthesis and development of the aragonite polymorphic form cockle shell-derived calcium carbonate nanoparticle (CSCaCO$_3$NP) of biogenic origin for a targeted drug delivery system [17]. This marine species product is known to possess good quality and pure crystalline CaCO$_3$ in purely aragonite polymorph form [77,78] more so, heavy metals such as arsenic, lead or mercury are absent in the shells [12,106,107].

Presently, CSCaCO$_3$NP has remarkably drawn the attention of many researchers due to its application as a drug nanocarrier for varieties of drugs and food supplements as a result of its unique, non-hazardous, specific and beneficial properties [15,91,97,108]. Further, studies demonstrated high biocompatibility of CSCaCO$_3$NP [10,16,22,97]. In addition, the biological safety evaluation of porous CSCaCO$_3$NP on hFOB 1.19 cells showed that CSCaCO$_3$NP could be a promising drug nanocarrier with wide safety level [16,25]. Danmaigoro et al. [17] and Ghafar et al. [106] claimed that CaCO$_3$NP of biogenic cockle shell origin have relatively large loading content capacity, encapsulation efficiency, functional group endings for electrostatic ion bonds (hydroxyl and carboxylic groups), surface structural porosity, higher surface area and response to pH degradation which explains its uniqueness as a novel inorganic compound for nanocarrier application. It was reported to exhibit great promising potential as a drug nanocarrier, these have become possible due to its immense biomedical utilizations attributable to its biocompatibility, osteoconductivity, pH sensitivity, hydrophilic nature as well as slow biodegradability [9]. In respect to that, the biogenic CSCaCO$_3$ has great characteristics to be a potential drug nanocarrier among others [10]. Great nanocarriers with homogenous shapes and size, surface morphology as well as distinctive compositions are derived to promote drug-system delivery applications notwithstanding the fact that homogenous spherically shaped CSCaCO$_3$NP aragonite nanoparticles among others, has a promising great impact in nanomedicine due to its ability to enhance the therapeutic efficacy and pharmacokinetics of drugs [17]. In the past few decades, continuous synthesis and development of different nanoparticles has been in process in the quest to elicit the dose effect of drugs at a low concentration with minimal toxic effect [2,6,18,21,85,94,95]. These diverse nanoparticles facilitate several mechanisms of action of drugs in the targeted area of the body system. Nevertheless, the limitations of these nanoparticles are becoming worrisome in nanomedicine therefore, the need of certain improvements, modifications and newly synthesised nanoparticles with good efficiency, standard quality and unique characteristics traverses across the field of nanomedicine [109,110]. CSCaCO$_3$NP was proven to be a pH sensitive drug delivery systems thus, drug release tends to be faster in a weak acidic environment (pH 4.8) than the normal environmental pH (7.4) [10]. However, the fate of this CSCaCO$_3$NP in a highly acidic environment is unknown.

### 2.4. Synthesis of CaCO₃ Aragonite from Cockle Shell

Several methods of synthesizing nanoparticles were developed, adopted and improved to modify the properties and minimise cost of production. In addition, several efforts were also made to improve the old existing methods to obtain more desired qualities of the nanoparticles through enhanced physicochemical properties [111]. Aragonite exhibits unstable thermodynamic properties while its size and shape produced in the laboratories are mostly dependent on the condition and methods used for its preparation [11]. From the early stage of micron sized CSCaCO₃NP fabrication up to the recent synthesis of desirable nanosized particles, many studies adopted the bottom-up approach through the precipitation process, either by carbonation [99] or solution methods [102]. The carbonation method is sometimes faced with difficulties in the control of the shape and surface modifications [112]. Further, the bottom-up methods also failed to produce appropriate sizes and shapes required for pure aragonite nanoparticles [98,99], because it is often in combination with other polymorphs e.g., calcite and vaterite [113]. Other bottom-up methods which were initially thought to be useful and productive in industries and also considered to be environmentally friendly, were later found to be quite tedious, expensive, time consuming and complicated. For example, the carbonation method is associated with strict temperature regulations, strenuous gas bubbling phase (mixture of $CO_2$ and $N_2$ or $CO_2$ alone) and the use of purified raw materials. Another limitation of the bottom-up method is that various toxic impurities are added to the final products which is toxic to normal cells [103]. The use of chemical additives in the bottom-up approach for the production of crystalline CaCO₃NP has a disadvantage because it affects the final chemical composition of the nanoparticle produced [98]. An example of such a method is the formation of needle-like crystalline vaterite CaCO₃NP in the presence of gold nanoparticles and $Mg^{2+}$ and the formation of calcium carbonate amplified with organic thiols, fatty acids and polymer [114,115], thus this combination is usually not suitable for biomedical use due to the associated non-biocompatibility and toxicity reports [116]. Hence, the bottom-up approach is regarded not appropriate for some specific biomedical research.

Scientists from different fields aspiring to synthesise great quality aragonite nanoparticles with biomedical importance have adapted the top-down approach with continuous modifications and improvements, which recently attracted attention due to the tremendous striking properties of aragonite nanoparticles. As synthesis of CaCO₃-aragonite from cockle shells are mainly performed using top-down (destructive) method perhaps this method allows CaCO₃-aragonite to be obtained in their natural forms by retaining and protecting their great unique features [11] since the method involves diminution of bulk material to micron size and finally to the desired nanoscale size particles [40]. Mechanically synthesized CaCO₃-aragonite nanoparticles have superior advantages over the chemically synthesized material since they are more biocompatible with cells with little or no toxic and immunogenic effect as earlier reported in previous studies [10,20,91]. This promotes their wide biomedical applications such as; drug delivery as a carrier, in diagnostic imaging as demonstrated by Kiranda et al. [22] and as a scaffold in bone tissue remodelling by Mahmood et al. [117]. Mechanical milling among all types of top-down method is the most extensive and widely used approach for the production of various nanoparticles, others are laser ablation, sputtering, nanolithography, and thermal decomposition [40].

Directing to the recent trend of synthesizing the CSCaCO₃NP, the early synthesis of CSCaCO₃NP was governed and supported few years back by the pioneering fundamental study of Islam et al. [101], who introduced a basic preparation procedure in transforming the solid structure of cockle shells into a pure nano-sized powder by developing a novel method through the use of simple mechanical stirring in the presence of BS-12 surfactant (dodecyl dimethyl betaine) which is a bio-mineralization material that aids smaller sized formation of nanoparticles. Later, a minimum procedural steps was adopted with no addition of impurities and was performed under room temperature [15]. The advanced development on the production of CSCaCO₃NP has led to the use of a high pressure homogenizer technique via microemulsion system in the presence of polysorbate 80 surfactant (Tween 80) to obtain a modified homogenous nanosized particles Kamba et al. [91]. Although, the technique demonstrated

promising results for drug delivery system, it had considerable setbacks such as the use of expensive and complex equipment with high operation energy input. Therefore, the top-down method adopted by the pioneering researchers with subsequent improvements that led to the better spherical size obtained recently, appeared to be a more appealing and convenient option for the synthesis of CSCaCO$_3$NP at large scale production. Hence, the method is simple at low cost and environmentally friendly, which involves little procedural steps without stringent temperature management. In addition, expensive hazardous chemicals or any carbonation methods and time-consuming gas bubbling are exempted. The mechanical grinding type of top-down method adopted, involves a suitable readily available and more cost-effective instrument in the presence of BS-12 surfactant, which is a biomineralization catalyst. Previous work reported rod-shaped nanoparticles for CSCaCO$_3$NP [91,101], modifications and improvement has led to a successful formation of homogenous spherically shaped CSCaCO$_3$NP [10,11,17,18,106]. An overview of the synthesis and characterization of CSCaCO$_3$NP is shown in Figure 2.

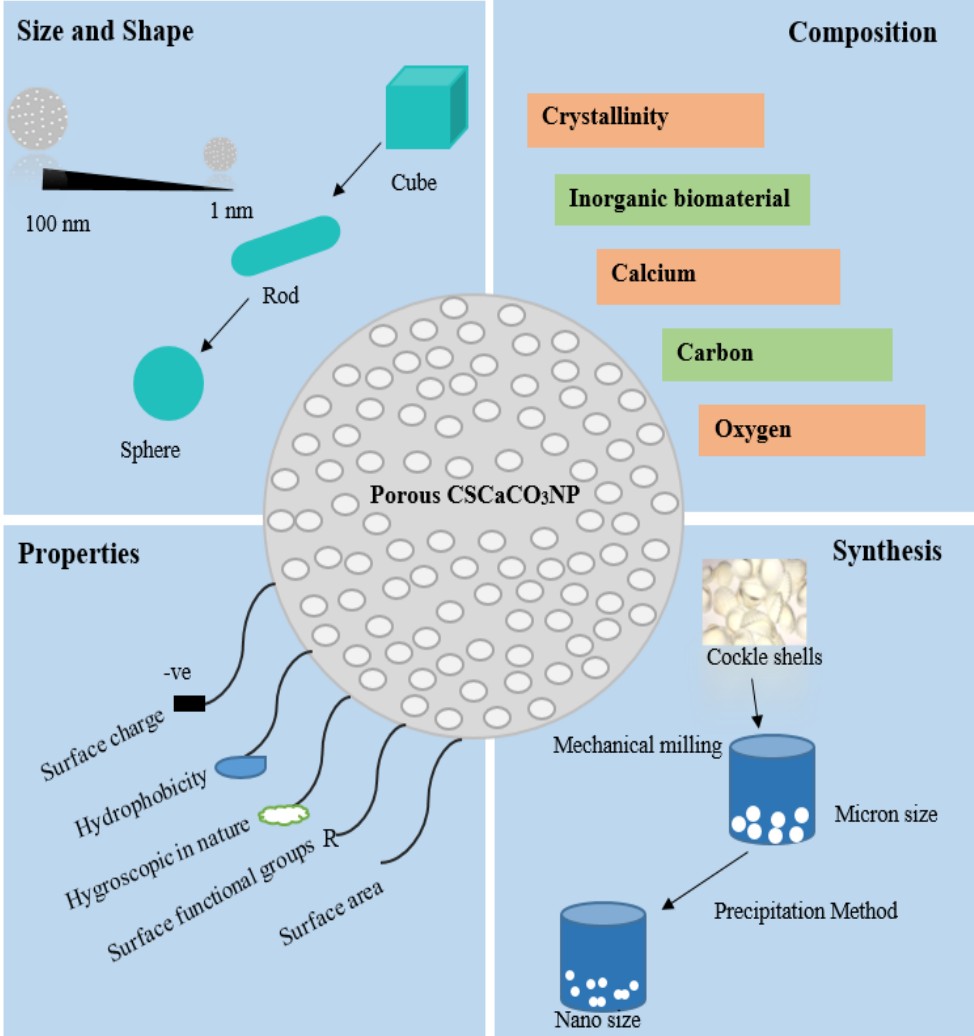

**Figure 2.** Properties and synthesis of veteran cockle shell-derived calcium carbonate nanoparticles.

*2.5. Characterization and Physicochemical Properties of CSCaCO$_3$NP*

(a)    Particulate Size and Surface Morphology

The particle size of nanoparticle is well known to be best characterized using transmission electron microscope and field electron microscope as these tools give the actual reliable geometrics [91]. Although

TEM operates with a different principle to the field emission scanning electron microscope (FE-SEM), they both give similar data. The two methods aim at providing evidence of the physicochemical properties of nanoparticles such as size, size distribution, porosity, shapes and cross sectional surface morphology [29]. It is well documented that a favourable range of average sized nanoparticles for oral absorption was beneficial and reliable to proleptic tissue distribution and passive targeting capabilities of that nanomaterial in vivo [23,45]. Previous literature documented that the particle size within the range of 10–100 nm are regarded as an optimum for nuclear and cellular uptake in smooth muscle and epithelial cells hence suitable for oral delivery of therapeutics [50]. Thus, biodistribution and cellular uptake of the encapsulated candidate drug is strongly associated with the size of a nanomaterial. Intravenous therapeutic delivery are restricted within the range limit of 20–150 nm which is considered suitable for the passage of nanoparticles at the lumen of the micro-capillary with a diameter of 200 nm [24]. In respect to that, a study conducted by Kamba et al. [118] reported a porous rod shaped $CSCaCO_3NP$ with an averaged particles size of 100 nm. In the contrary, Fu et al. [10] demonstrated an average size ranging between 20–60 nm for $CSCaCO_3NP$ with a porous spherically shape morphology hence favourable for drug loading and encapsulation. Generally, the size of nanoparticles has a profound effect during release of the encapsulated materials. In addition, particle size increases when measured with zeta nanosizer mainly due to the wavelength dynamic principles and Brownian motion of the suspended nanoparticles in solution as compared to the real time measurement of the size of the nanoparticle using TEM which usually causes shrinkage of the nanoparticles as a result of the coated copper grid during sample preparation [119]. The smaller the particles, the larger the surface area which in turn promotes drug loading. Contrary to that, slow diffusion within the inner core shell of larger sized nanoparticles takes place [29,120]. Mutual compromise exists between the size of the nanoparticles and storage stability [29]. Small sized nanoparticles agglomerate over time during storage compared to large sized nanoparticles. The longer the storage period the more they become agglomerated [120].

(b)  Surface Charge and Stability

The surface charge characterization of nanoparticles is essential for the determination of its interaction with the target site in vivo. The intensity and surface charge depicts particulate interaction with the biological system as well as their interaction with the active biological compounds [29]. Although, zeta potential measurements are sometimes used to characterize the size of nanoparticles in colloidal form, it is best used in detecting the surface charges, which could be either positive or negative charged zeta potentials thus, confirms the stability nature of the particles. Polydispersity gives a detailed explanation of how stable a nanoparticle is in a colloidal solvent (uni-dispersity) as all the particles will be in a random uniform motion [17]. The average high zeta potential above ±30 mV is capable of causing strong electrostatic repulsion as such prevents aggregation of the nanoparticles and thus is considered a good stable range [121]. Thus the higher the potential charge, the greater the electrical repellent forces which are likely to hinder the agglomeration of the nanoparticles [122]. Although, uptake of nanoparticles is concordant to increase in +/− charge ratio of the nanoparticles, an excessive positive charge can induce toxicity which is likely to cause unwanted immunological responses in the system [110]. Encapsulated candidate drugs might experience a prolonged circulation period in the body when administered, since the surface charge of the nanoparticle is negative [123]. Candidate drug properties and the variation in the methods of nanoparticles synthesis greatly affects the negative charge potential of $CSCaCO_3NP$. Thus, $CSCaCO_3NP$ has so far been reported to exhibit a negative zeta potential with a decrease or increase in charge value after drug loading as shown in Table 1.

**Table 1.** Variations in the negative potential charge exhibited by cockle shell-derived calcium carbonate nanoparticles (CSCaCO$_3$NPs) before and after loading with different drugs.

| Author's Name | Drug Used | Zeta Potential (mV) before Drug Loading | Zeta Potential (mV) after Drug Loading | Size of Nanoparticles (nm) | Shape of Nanoparticles |
|---|---|---|---|---|---|
| Isa et al. [28] | Ciprofloxacin | −15.3 ± 2.0 | −13.0 ± 1.9 | 11.93–22.12 | Spherical |
| Fu et al. [15] | Doxorubicin | −46.17 ± 3.82 | −40.57 ± 3.80 | 20–60 | Spherical |
| Jaji et al. [16] | Teriparatide (PTH 1–34) | Nil | −27.6 ± 8.9 | 30 ± 50 | Spherical |
| Hammadi et al. [105] | Taxanes (Docetaxel) | −15.4 ± 0.9 | −21.7 ± 0.1 | 42.22 | Spherical |
| Danmaigoro et al. [32] | Doxorubicin | −12.1 | −34.7 | 24.90 | Spherical |
| Ghafar et al. [6] | Nil | −9.4 ± 0.8 | −28.9 ± 0.1 | <100 | Spherical |
| Kiranda et al. [82] | Nil | Nil | −16.4 ± 3.81 | 35 ± 16 | Spherical |
| Hamidu et al. [20] | Doxorubicin | −19.1 ± 3.9 | −17.8 ± 4.6 | 35.50 | Spherical |
| Ghaji et al. [124] | Cytarabine | −11 | −13.2 | 20–50 | Spherical |

(c)   Functional Groups and Conjugation

FTIR is a special technique for detecting the functional groups at different phases of inorganic and organic compounds. It is an important guideline for elemental characterization of CaCO$_3$ phases associated with different peaks depicting dissimilarities in the constituents of carbonate ions [11]. This important tool also reveals the fingerprints regions of the conjugal drugs as well as presence of impurities incorporated in the nanoparticles, this technique is very important for nanomaterials used in the drug delivery system [17]. Raju et al. [125] stated that the carbonate ions and molecules that have similarities with carbonate ion compounds generally have vibrational peak modes of v1–v4 that are basically four in number; out-of-plane bending, symmetric stretching, doubly degenerate planer bending and doubly degenerate asymmetric stretching. Raju et al. [112] described the peaks of CO$_3{}^{2-}$ to be similar with the v1–v4 vibrations with little changes on the peaks after loading process, which is an illustration of the shift on the vibrational peaks in the milieu of oxygen atoms and the modified electrostatic valence force that exists in Ca–O bond. Thus, the characteristic band spectrum of CSCaCO$_3$NP usually does not reveal any critical shift after loading with drugs compared to the peaks of the original CSCaCO$_3$NP. In support of that, other scientists observed negligible shifts of aragonite bands after loading with doxorubicin [10,11,15,17]. The stretching frequency decreases with increase in the bond formation of CSCaCO$_3$NP with any candidate drug, this is attributable to the slight shift of the wavelength from higher to lower region of the frequency [126]. In addition, the peak at 1082 cm$^{-1}$ indicates a distinct sharp wavelength presence on the characteristic aragonite phase of CaCO$_3$ spectrum of which the ions are vague at the infrared region [11,125].

(d)   Crystallographic and Purity Nature of CSCaCO$_3$NP

Crystallinity and purity nature of CaCO$_3$NP is mostly analysed using X-ray diffraction which is considered a strong tool used in assessing the purity nature and crystalline content by identifying the arrangement of atoms and molecules in their crystal or amorphous states [17]. Its provides standard techniques for assessing the crystallinity nature at atomic scale which gives reference guidelines in identifying CaCO$_3$ polymorphic nature using their intensity and position as fingerprints [11,127]. Presence of less or negligible changes on the crystal nature of aragonite phase after loading indicates a successful drug loading without any impurity as reported by Fu et al. [10], Kamba et al. [91]. However, Kamba et al. [91], Isa et al. [97] and Hammadi et al. [18] reported the prominent peaks of CSCaCO$_3$NP at 2 theta positions of 26.5°, 27° and 33.3° for CSCaCO$_3$NP.

(e)   Porosimetry

Determination of the surface area to volume ratio of a nanoparticles is essential for their characterization considering the titanic influence of the surface area on the properties and performance of the nanoparticles both in vivo and in vitro [40]. Nitrogen adsorption and desorption isotherm is use for the determination of nanomaterial porosimetry in order to estimate the pore volume, specific surface area and pore size distribution [128]. Danmaigoro et al. [17] and Hammadi et al. [18] reported

a specific surface area for CSCaCO$_3$NP to be 6.181 m$^2$/g and 6.9529 m$^2$/g respectively and further categorised the CSCaCO$_3$NP curve of the Brunauer–Emmet–Teller (BET) as a Type III with hysteresis loop indicating multiple layers based on the classification of adsorption–desorption isotherm.

### 2.6. Safety of CaCO$_3$ Aragonite Nanoparticles

The administration of nanoparticles in vivo is puzzled by problems related to bioaccumulation, toxicity and biodistribution although propounding studies are ongoing to overcome these challenges [129]. Toxicology deals with the monitoring of arrays of sequential events with regards to metabolism, distribution, progress and acquaintance of a material for possible adverse effect within the system [130]. A new branch of toxicology known as nanotoxicology is mainly concerned with the study of the properties and toxicity of nanomaterials with regards to their possible deleterious effects by recommending a concise test protocols on their safety assessment [130,131]. The objective behind scientific research in the field of medicine, pharmacy and other health related disciplines is the development of controlled delivery systems with less or no toxic effect because most drugs have some associated side effects, less selectivity as well as non-specificity when taken at high dose [91]. Generally, the toxicity and safety of CSCaCO$_3$NP is directly dependent on its physicochemical nature with special emphasis on the particulate size [16]. Biocompatibility studies of blank CSCaCO$_3$NP presented a high percentage viability (>80%) even at high concentration of the nanoparticles which shows its toxic free nature [10,11]. Great efforts have been devoted to the study of CSCaCO$_3$NP with promising therapeutic effects that holds a good future to nanomedicine with regard to its development as delivery system with negligible or no toxicity. The safety of subcutaneous doses of this biogenic nanoparticle was tested on rats and a dose of 59 mg/m$^2$ and 590 mg/m$^2$ was considered safe as it showed negligible mild lesion and no mortality was recorded but some clinical signs and histopathological conditions were reported at a high dose of 5900 mg/m$^2$ and 29,500 mg/m$^2$ [16]. In addition, a repeated intravenous administration of Dox-loaded CSCaCO$_3$NP in dogs revealed a cumulative dose of 150 mg/m$^2$ and below had no significant toxic effect suggesting a good safety margin of CSCaCO$_3$NP-Dox at a dose of 30 mg/m$^2$ [21]. In addition, previous literature documented the ability of CSCaCO$_3$NP to stimulate cellular differentiation leading to an increase in growth of osteoblast cells [11,132].

### 2.7. Drug Delivery System

Drug delivery could be via intramuscular, oral, intravenous or subcutaneous routes which helps in the distribution of drugs in the body system mainly by diffusion, absorption and blood circulation which results in the accumulation of a reasonable concentration of the amount of drug needed by the body at a specific targeted area to achieve the therapeutic effect [2,33]. Nanocarriers administered via the oral route are believed to be highly effective due the fact that oral route administration of drugs are considered the most desirable delivery system because its presents the most readily acceptable, non-invasive and convenient alternative [133]. Varieties of abundant delivery systems exists which are mostly organic and some available inorganic ones that have been discovered and synthesized recently in modern pharmaceuticals and nanomedicine areas [6]. The inorganic materials include calcium phosphate, tri-calcium phosphate, hydroxyl apatite, calcium carbonate [134], silicon, iron oxide, layered double hydroxide and colloidal gold [9]. Drug delivery systems are known to enhance efficacy thereby masking some adverse effects in the body [135]. In addition, a controlled drug release and targeted delivery should be concordant to overcome the challenge of the sustained release [111,121]. Distribution of a drug within delivery systems to a specific targeted bio-site in a controlled and sustained pattern are reported as an efficient drug delivery leading to an increased bioavailability and intrinsic activity of the therapeutic candidate drug [136]. CSCaCO$_3$NP has a longer biodegradation time, hence, administered candidate drugs can be retained for a long duration of time before the final release at the specific site of action [10].

### 2.7.1. Drug Loading Efficiency and Encapsulation Efficiency

Loading efficiency and encapsulation efficiency are of paramount importance in nanomedicine, these two parameters refer to the mass ratio of drugs to nanoparticles and utilized drugs (amount of drug fed) after the drug loading process respectively [2]. However, nature, charges, structures, and physicochemical properties of the nanocarrier materials influences the loading efficiency, likewise mass of the drug fed, drug solubility, structural affinity of the drug, mechanism of drug-loading and other experimental conditions determine the final encapsulation efficiency [137,138]. In the most part, high loading efficiency is difficult to get compared to high encapsulation efficiency for some nanoparticles, although nanoparticles with high loading content are obtained using methods with high encapsulation efficiency. Moreover, crystallization, dipole interaction, physical attraction, precipitation, covalent bonds, coordinate bonds and hydrogen bonds result in high encapsulation within the matrix of nanoparticles [2,139]. Despite the general significant increase in the size of the nanoparticles after loading as reported by a considerable number of researchers [18,22,140,141], it has been observed that the drug loading and encapsulation do not have any significant effect on the morphology of the nanoparticles. Hence, the shape and nature remains unaltered while the size increases after loading depending on the existing bond interactions between the nanoparticles and the type of drug loaded [11,17]. Drug molecules consolidation onto a nanoparticle occurs during nanocarrier synthesis followed by adsorption of the drug leading to drug incubation within the nanocarrier [142]. Thus CSCaCO$_3$NP is capable of loading drug molecules in large quantity due to its large surface area, functional group endings and porosity through the process of adsorption [143]. The negatively charged CSCaCO$_3$NP usually demonstrates minimal drug lost during the loading process [10]. A study explained the high loading content of CSCaCO$_3$NP to be attributable to its charge potential, therefore positively charged drugs could be efficiently encapsulated since the electrostatic interactions between the different molecules was amplified [10]. Hammadi et al. [18] reported a decrease in the percentage entrapment by CSCaCO$_3$NP as the amount of drug decreases and vice versa although, this is not always so as the condition varies with the type of drugs used.

### 2.7.2. Mechanism of Release Action of Drug from CSCaCO$_3$NPs

Evaluation of the kinetic release of encapsulated drugs from CSCaCO$_3$NPs is the fundamental basis for understanding the release mechanism. The different release pattern of a drug from a nanocarrier could either be monophasic, biphasic or tri-phasic [144]. The biphasic release pattern is mostly demonstrated by inorganic nanocarriers which involves the rapid initial burst release at the first phase which is strongly associated with excess surface attached drug molecules on the nanoparticles as a result of the weak bonding [145]. The second phase of the biphasic release involves the consistent slow and sustained release reaching the plateau phase [146]. The kinetic release mechanism of a nanocarrier is strongly influenced by the nature of the candidate drug loaded, nature of the nanomaterial itself and nature of conjugation process which could either be as a result of the ionic interaction or physical entrapment of the drug molecules and its microenvironment [147,148]. Thus, the decomposition, diffusion rate, solubility and degradation of the candidate drugs also have a strong role to play during the release of the drug molecule from the matrix of the nanocarrier which could either lead to a sustained and controlled release or rapid release [149]. However, fully encapsulated drugs within the nanocarrier regardless of the existing bond, are released from the carrier system by either disintegration, pH of the microenvironment or degradation of the nanocarrier which eventually leads to breakage of the existing bond holding the drugs within the matrix [150]. Temperature, enzymes and pH from the microenvironment usually initiate or trigger the release mechanism of the nanocarrier within the body [151]. The release mechanism of nanocarriers triggered by pH could be as a result of sensitivity of the functional group endings to pH changes; examples of these functional group endings on nanocarriers are sulphonamide group, ammonium salt component, carboxylic group etc. [152]. The carrier molecules undergo different processes of drug release either by dissolution, carrier matrix degradation or breakage of bonds between the nanocarrier and the drug molecules [153]. However,

calcium carbonate undergoes degradation through the process of diffusion to release its entrapped matrix contents [154].

The release of encapsulated drugs from CSCaCO$_3$NP is basically pH dependent, which is more responsive to a weak acidic environment than normal physiologic environment and vice versa depending on the type of drug loaded. Thus CSCaCO$_3$NP could be called an excellent pH-responsive carrier [98]. A study conducted by Fu et al. [10] suggested that CSCaCO$_3$NP loaded with doxorubicin, administered for the purpose of chemotherapy could hardly release its encapsulated materials in the blood stream and normal cells before reaching it targeted area. An effective drug delivery system is said to have achieved an excellent sustained and controlled release phase to liberate its matrix content (drug molecules) at a specific localised site for effective therapeutic purposes [155]. In addition, the regulation of adequate therapeutic blood level of drugs at the minimum effective concentration and the minimum toxic concentration is achieved by the controlled kinetic release [153]. It was documented that CSCaCO$_3$NP slowly decomposes in a physiologic state to produce Ca$^{2+}$ and CO$_2$$^{3-}$ thereby releasing the encapsulated material from its inner matrix of which the by-products could be useful in bone remodelling, blood clotting and muscle contraction [21]. However, the exact fate of the control release mechanism of CaCO$_3$ of biogenic cockle shells origin in low pH remains unknown and yet to be documented.

## 2.8. Therapeutic Application of CSCaCO$_3$ Aragonite Nanoparticles in Drug Delivery

Efficient and safe delivery systems which are capable of delivering therapeutic agents by enhancing their solubility and bioavailability to molecular or sub-cellular levels of infected cells in any diseased condition without harming normal healthy cells are of paramount importance for researchers in the field of nanomedicine [156]. Manifold trials and successes were seen in the delivery systems for therapeutics agents ranging from soluble to insoluble drugs, yet some limitations regarding biodegradability of such materials, stability in serum plasma level, inconsistent degradation of toxic by-products and high toxicity, which is mostly size and shape dependent, have prevented their smooth achievable clinical acceptance [17].

The application of CSCaCO$_3$NP in bone remodelling and osteoporosis therapy was documented in the previous work of Jaji et al. [11] who observed the safety and proliferation effects of CSCaCO$_3$NP on osteoblast cells and concluded that CSCaCO$_3$NP is a cytocompatible and antiosteoporotic agent. Kamba et al. [19] reported a positive outcome of the cytotoxicity and genotoxicity analysis of CSCaCO$_3$NP on normal fibroblast cells, thus, a negligible increase in ROS generation after exposure of normal fibroblast cells to CSCaCO$_3$NP at a dose of 100 μg/mL with a slight increase of ROS at a higher dose of 200–400 μg/mL after 72 h were observed. CSCaCO$_3$NP as a bone remodelling agent was also supported and proven in previous literature [104,143].

CSCaCO$_3$NP can be used in cancer research therapy. The therapeutic efficacy of Docetaxel loaded CSCaCO$_3$NP demonstrated 22% viability on breast cancer cells (MCF-7) indicating an interesting high level of cytotoxic effects on cancer cells [18]. Purely CSCaCO$_3$NPs were assessed based on their pH sensitivity for effective delivery of doxorubicin against a highly malignant primary bone cancer (osteosarcoma) [10]. In addition, studies reported improved therapeutic effects of anticancer drugs encapsulated with CSCaCO$_3$NP for the treatment of osteosarcoma, breast cancer, bone tumours and acute myelogenous leukaemia [26,84,103,113].

The application of CSCaCO$_3$NPs in nano-antibacterial therapy was successful and effective as reported in the previous work of Isa et al. [97], who observed an enhanced antibacterial efficacy of ciprofloxacin encapsulated with CSCaCO$_3$NP and its biocompatible effect on macrophages cell lines. In addition, CSCaCO$_3$NPs were reported to efficiently encapsulate and boost the potency of vancomycin, hence, they can serve as a nanoantibiotic bone implant for the treatment of osteomyelitis [104].

However, due to the hard and slow biodegradable nature of cockle shells, a study demonstrated the use of CSCaCO$_3$ powder particles as an alternative construction material for artificial reef which significantly improved the strength of the artificial reef [157].

### 3. Conclusions

Veteran CSCaCO$_3$NP has recently been synthesized and developed by quite a number of scholars in the quest to overcome some biocompatibility issues associated with some therapeutic agents. The CSCaCO$_3$NP among other nanoparticles, is one of the potential and toxic free nanoparticles due to its unique properties such as biodegradability, bioavailability, biocompatibility, large surface area and porosity nature. The top-down method adopted for the synthesis of CSCaCO$_3$NP involves the use of simple suitable available and more cost-effective instruments in the presence of BS-12 surfactant, which is a biomineralization catalyst. CSCaCO$_3$NP is applicable for the delivery of anticancer and antibiotics drugs with a remarkably effective role in bone remodelling and osteoporosis therapy. Nevertheless, the slow biodegradable nature of powdered cockle shells qualified its usage as an alternative construction material for artificial reef. Previous research literature highlighted some advantages of CSCaCO$_3$NP as a nanocarrier for anti-cancer drugs and antibiotic focusing more on intravenous administration. Hence, the current review recommends more research on kinetic release mechanism of CSCaCO$_3$NP in various different pH with emphasis on high acidic pH. Further improvement and modifications on the method of synthesising CSCaCO$_3$NP to prevent problems of agglomerations is expected. Finally, research focus should be made on CSCaCO$_3$NP as carrier system for various therapeutic agents such as anti-oxidants, anti-inflammatories using different routes of administration for effective applications at large.

**Author Contributions:** All authors contributed to the drafting of this manuscript. Conceptualization, M.M.M.; M.A.M.M. and Z.A.B.Z. motivation, M.A.M.M.; writing—original review, M.M.M. and K.A.; editing manuscript, Z.A.B.Z.; M.A.M.M.; E.B.A.R.; M.M.M.; A.D. and S.M.C. All authors approved the final draft of the manuscript.

**Funding:** The work was financially supported by the Universiti Putra Malaysia (Grant number GP-IPS 9663600).

**Conflicts of Interest:** The authors declare no conflict of interest.

### Abbreviations

| | |
|---|---|
| CSCaCO$_3$NP | Cockle shell-derived calcium carbonate nanoparticles |
| CaCO$_3$ | Calcium carbonate |
| BET | Brunauer–Emmet–Teller |
| TEM | Transmission electron microscope |
| MPNs | Metal phenolic nanoparticles |
| MOFs | Metal organic frameworks |
| FE-SEM | Field emission scanning electron microscope |
| BS-12 | Dodecyl dimethyl betaine |
| GIT | Gastrointestinal tracts |
| LUV | Large unilamellar vesicles |
| SUV | Small unilamellar vesicles |
| MLV | Multilamellar vesicles |
| ROS | Reactive oxygen species |

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
