# Peer review of "Cockle Shell-Derived Calcium Carbonate (Aragonite) Nanoparticles: A Dynamite to Nanomedicine"

_applsci, doi:10.3390/app9142897_

Round 1

Reviewer 1 Report

The review “Cockle Shell-Derived Calcium Carbonate (Aragonite) Nanoparticles: A Dynamite to Nanomedicine” highlights recent achievements in production and utilisation of cockle shell-derived aragonite for drug delivery and other bioapplications.

This review is of current interest in the field of biotechnology and drug delivery and systematically touches on major aspects of the use of cockle shell-derived CaCO3 – strategies for production; physical and chemical properties that are important for drug encapsulation and controlled release; advantages and limitations for therapeutic applications with the focus on toxicity and biocompatibility issues; and current and potential obstacles in this field. The manuscript could be improved as follows:

1. Advantages of aragonite over other polymorphs of CaCO3 should be highlighted (especially vaterite that also possesses high porosity, even higher than aragonite; biocompatibility and low or no toxicity; and is of spherical shape by its nature, although this can be modified to rod-like and other shapes if necessary).

2. Comparison of the production strategies is poorly described or sometimes missing, e.g. what makes CS-derived CaCO3 superior over aragonite chemically synthetized under laboratory conditions?

3. The authors highlight the biocompatibility of CSCaCO3 nanoparticles several times making the impression that this is one of the key features of these nanoparticles that makes them favourable for drug delivery applications. However, nearly all types of nanoparticles mentioned in the second section of this review are also biocompatible and non-toxic, so I suggest highlighting high porosity and availability of surface modification approaches rather than biocompatibility.

4. Row 324 : 98% of the shells is CaCO3. Is it purely aragonite polymorph or the mixture? Please, specify.

 5. Table 1: It make sense to add the sizes of NPs used, and their shapes if dirrerent from the needle-like.

Author Response

The authors will like to appreciate the efforts of the reviewer for the attention given in reviewing the entire manuscript. The critical comments and helpful suggestions were all taken into considerations by the authors. Based on the valuable comments, the review paper is greatly fixed and improved.

Reviewer 2 Report

The authors provide an interesting review on the use of cockle shell for making calcium carbonate nanoparticles for nanomedicine. Roughly 1/3 of the review focuses on other nanomedicine technologies, but I believe some work could be done to improve that. For example, there are essentially 2 lipid sections, which could be combined potentially. And there are very new technologies like carbon nanotubes, but missing other new technologies such as mofs or mpns, while also missing old technologies, such as polymer particles, gold nanoparticles, silica, etc.  The review shuoldn't focus too heavily on these, it just seemed that the choice of seciton wasn't fully representative.

The sections on caco3 are interesting and informative, and I did find the historical and geographical aspects interesting, though I can't guarantee every reviewer will find that interesting. The field is still fairly small, but I think that makes this review timely as it will help the field to grow.

The spelling, grammar, tenses, etc, need a significant overhaul, and even simple things like CaCO3 vs CaCO3 aren't consistent. So I recommend the authors to employ a professional service for this, as I think the review has merit, but in it's current state is hard to read comfortably.

In conclusion, it's informative and likely timely, but the non-caco3 portion needs a bit of a re-think and some focus, and the writing needs some work. Otherwise it should be published.

Author Response

The authors will like to appreciate the efforts of the reviewer for the critical observations, comments and helpful suggestions. The authors have taken all the comments and suggestions of the review into account and this have greatly improved the review paper.
